# Microbiological Survey and Antimicrobial Resistance of Foodborne Bacteria in Select Meat Products and Ethnic Food Products Procured from Food Desert Retail Outlets in Central Virginia, USA

**DOI:** 10.3390/pathogens12070965

**Published:** 2023-07-23

**Authors:** Chyer Kim, Brian Goodwyn, Sakinah Albukhaytan, Theresa Nartea, Eunice Ndegwa, Ramesh Dhakal

**Affiliations:** 1Agricultural Research Station, Virginia State University, 1 Hayden Drive, Petersburg, VA 23806, USA; endegwa@vsu.edu (E.N.); rdhakal@vsu.edu (R.D.); 2Department of Agriculture, Food, and Resource Sciences, University of Maryland Eastern Shore, Princess Anne, MD 21853, USA; bmgoodwyn@umes.edu; 3AlAhsa Health Cluster, Ministry of Health, Hofuf 36421, AlAhsa, Saudi Arabia; ssalbukhaytan@moh.gov.sa; 4Cooperative Extension, Virginia State University, 1 Hayden Drive, Petersburg, VA 23806, USA; tnartea@vsu.edu

**Keywords:** food desert, LCSMs, SIOMs, economic viability, packaging surface, ethnic food, foodborne pathogens, AMR, MDR

## Abstract

In food desert areas, low-income households without convenient transportation often shop at small, independently owned corner markets and convenience stores (SIOMs). Studies indicate a higher potential for reduced product quality and safety of foods sold at SIOMs, with more critical and non-critical code violations in the region. This study aimed to assess the difference in market scale on the microbiological quality in select food products procured from food deserts in Central Virginia. A total of 326 samples consisting of meat products (i.e., ground beef, chicken, and sausage), ethnic food products (i.e., ox tail, stock fish bite, egusi ground, and saffron powder), and food packaging surfaces procured from ten registered SIOMs and nine large chain supermarkets (LCSMs) between August 2018 and March 2020 were evaluated. Higher levels of aerobic mesophile and coliform counts were found in SIOMs-acquired samples than in LCSMs-acquired samples, as demonstrated by the lower food safety compliance rate of SIOMs. Regardless of SIOMs or LCSMs, *Campylobacter*, *E. coli*, *Listeria*, and *Salmonella* were detected in 3.6%, 20.9%, 5.5%, and 2.7% of samples, respectively. The majorities of *Campylobacter* (75%, 6/8) and *Salmonella* (83.3%, 5/6) detected were from SIOMs-acquired samples including ethnic food products. Among the tested antimicrobials, AMP (100%) and TOB (100%) showed the highest frequency of resistance among *Campylobacter*, TCY (69.9%) among *E. coli*, NAL (100%) among *Listeria*, and TCY (50%) among *Salmonella*, respectively. The prevalence of multi-drug resistance (MDR) and non-susceptibility in *Campylobacter* and non-susceptibility in *Listeria* isolated from SIOMs-acquired food products were lower than those isolated from LCSMs-acquired samples. A higher price of the same brand name commodity sold at SIOMs than those sold at LCSMs was also observed, indicating an increased financial burden to economically challenged residents in food desert areas, in addition to food safety concerns. Elaborated and in-depth research on a larger-scale sample size with a greater diversity of products is needed to determine and intervene in the cause(s) of the observed differences in the prevalence of the pathogens and AMR profiles.

## 1. Introduction

The U.S. Department of Agriculture reported that food deserts are home to 37 million Americans [1]. Typically in food desert areas, low-income households without convenient transportation often shop at small, independently owned corner markets and convenience stores (SIOMs). The lack of nearby supermarkets results in reliance on nearby SIOMs to access fresh foods (i.e., produce, meats, dairy, and eggs). The research found fewer chain stores and more SIOMs in low-income areas [2,3].

Products handled by SIOMs on a relatively small scale might be quite different from those handled by their large-scale counterparts, in that they are generally unregulated and may come with self-prescribed handling and sanitation practice procedures that do not correlate with governmental and industry regulatory guidance. Newsad et al. [4] reported that FDA certification at privately owned stores is lower than at corporate-owned grocery stores. Potentially, the lack of effective Good Handling Practices (GHP) at SIOMs opens food safety vulnerabilities with the eventual foodborne illness outbreak within low-income communities. Research examining SIOMs food quality indicated a reduced perishable food quality and safety potential [3,5,6,7].

Additionally, reports indicated that foods in food desert stores were fair to poor quality compared to those in non-food desert retailers [8], and SIOMs might have more critical and non-critical code violations in food safety [9]. However, food safety risks associated with local food outlets, particularly in food deserts, appear insufficiently addressed in research practice or the literature.

Unknowingly, consuming foods contaminated by bacterial foodborne pathogens contributes to millions getting sick and thousands losing their precious lives in the U.S. [10]. Annually, in the U.S., infections associated with antimicrobial-resistant (AMR) bacteria also negatively affect the well-being of millions of humans and animals [11]. Due to the potential differences in handling and manufacturing practices, any differences in the prevalence and AMR of potential foodborne pathogens from SIOMs and large chain supermarkets (LCSMs) operations are of interest. Knowledge gaps exist in unique food safety risks associated with different types/sizes of food producers and processors in food desert areas. With growing media attention on consumer food safety and local food marketing, this type of research effort will help address knowledge gaps for increasing positive health outcomes for at-risk, low-income populations that live in food deserts, and will also assist food outlet vendors and managers in at-risk, low-income communities to better comply with food safety guidelines and regulations in order to reduce the risk of foodborne illness for their consumers.

A review of the USDA Food Environment Atlas database [12] indicated that widespread areas of the cities of Petersburg and Colonial Heights in Virginia (VA) are food deserts, with over one-third of low-income residents driving 1–10 miles to shop at supermarkets. In addition, although approximately 1.7 million Virginians live in a food desert [13], a review of existing local foods and the food safety literature uncovers the deficiency of assessment data on the microbiological safety of food products sold at food outlets in food desert areas of Virginia. 

Therefore, this study was designed to assess the effect of management differences/market scale (SIOMs vs. LCSMs) on the microbiological quality, level of potential foodborne pathogens, and prevalence of AMR bacteria present in selected meat products and ethnic food products sold in and surrounding food desert areas of Petersburg and Colonial Heights, VA. Furthermore, the difference in microbial counts on the packaging surfaces of commonly found hot dogs and sausages at both SIOMs and LCSMs was evaluated to determine any correlation between GHP and the hygienic condition.

## 2. Materials and Methods

### 2.1. Food Safety Inspection

Following the checklist excerpted from the Virginia Beach Department of Public Health [14], Illinois McHenry County Department of Health [15], and South Dakota Department of Health [16] educational materials, visual inspections for market compliance with the simplified food establishment checklist were performed during sample acquisition visits at markets. In brief, the market conditions and employee hygiene practices for ten operational compliance elements associated with food safety were assessed. The ten elements were classified into two groups based on the guidance cited above: Food Storage Practices and Employee Practices. In this study, the assessment of food storage practices were limited to packaging, storage area, and holding temperature, whereas employee practices were limited to personal clothing and hygiene. Packaging was further assessed for the following indicators: clear indication of expiration or sell-by date, visual observation of spoilage and mold, and external leakage of fluids (e.g., blood). Storage area was assessed for the following indicators: clear separation of raw meat from produce, no evidence of pests, and cleanness of surfaces. Holding temperature was assessed for the following indicators: refrigerator at or below 5 °C, freezer at or below −18 °C. Employee Practices were assessed by observing the following indicators: if wearing a hair net, gloves, uniform or at least no street clothing, and no consumption of food in product storage and preparation areas.

### 2.2. Expiration Date/Use-by-Date and Price

For the comparison of expiration date/use-by date and price, a total of 281 food items consisted of 148 and 133 samples from registered nine LCSMs and ten SIOMs, respectively, located in food desert areas of Petersburg and Colonial Heights in Virginia between August 2018 and March 2020 were assessed. Sample numbers are different due to the limited variety of the same type of products available at each store. Products assessed represented five different types of meat products. Additionally, for precise comparison, the same brand name of sausages and hot dogs sold at both LCSMs and SIOMs were evaluated. In this study, the term “Expiration date” or “Use-by date” shown on the product packaging is used for the comparison and is defined as “use-by date” for illustration purposes. The date is calculated either back (−, days before use-by date) or forward (+, days past use-by date) from the date when the sample was procured from the store. The price of each item was presented excluding tax.

### 2.3. Microbial Testing

Sample preparation and microbial analysis were conducted following standard FDA methods [17]. This study was divided into three sections. Section I was conducted on a total of 220 meat samples, which consisted of bacon, beef cuts, ground beef, chicken leg, chicken wing, pork chop, and pork sausage procured from nine LCSMs and ten SIOMs. To open sealed packages for sampling, one corner of each bag was sprayed with 70% ethanol, air-dried and then cut with flame-sterilized scissors. For blendable products (i.e., bacon, beef cut, ground beef, and pork sausage), approximately 25 g of each sample portion (taken from multiple locations in a sample) was homogenized with 225 mL of sterile 0.1% peptone water (PW) in a laboratory blender (Model 400 Circulator, Seward, Ltd., West Sussex, UK) at 260 rpm for 2 min. For other non-blendable products (i.e., chicken leg, chicken wing, and pork chop), each whole sample was aseptically transferred into a stomacher bag filled with an equal weight of sterile PW. Each sample was then agitated and vigorously rubbed by a gloved hand for 2 min to detach bacteria.

Section II was conducted on a total of 61 sausage and hot dog packaging surfaces obtained from nine LCSMs and nine SIOMs. As described in the non-blendable products above, each sample with the packaging was aseptically transferred into a stomacher bag filled with the equal volume of sterile PW and then agitated and vigorously rubbed by a gloved hand for 2 min to detach bacteria from the packaging surface.

Section III was conducted on 45 ethnic food products obtained from eight SIOMs. As described above, to open sealed packages for sampling, one corner of each bag was sprayed with 70% ethanol, air-dried and then cut with flame-sterilized scissors. For blendable products (i.e., fat back, seasoned meat, pork jowl, pollock fillets, seafood salad, ground shrimp, ground egusi, dried okazi leaves, and saffron powder), approximately 25 g of each sample portion (taken from multiple locations in a sample) was homogenized with 225 mL of sterile 0.1% PW in a laboratory blender (Model 400 Circulator, Seward, Ltd., West Sussex, UK) at 260 rpm for 2 min. For other non-blendable products (i.e., chicken, cow feet, goat meat, lamb meat, ox tail, pork BBQ, smoked anchovies, smoked shrimp, stock fish bites, and date), each whole sample was aseptically transferred into a stomacher bag filled with an equal weight of sterile PW. Each sample was then agitated and vigorously rubbed by a gloved hand for 2 min to detach bacteria. 

All samples were purchased in duplicate from each of the registered market sources located in food desert areas of Petersburg and Colonial Heights in Virginia between August 2018 and March 2020. Purchased samples were transported to our laboratory in insulated containers packed with ice. All products were kept in the refrigerator (4 ± 2 °C) and used for microbial testing within 24 h of the arrival.

Appropriate dilutions of the homogenate were surface plated with a detection limit of 100 cells per g (2 log CFU/g of sample) for blendable products (i.e., fat back, seasoned meat, pork jowl, pollock fillets, seafood salad, ground shrimp, ground egusi, dried okazi leaves, and saffron powder), and 20 cells per ml (1.3 log CFU/ml of wash fluid) for non-blendable products (i.e., chicken, cow feet, goat meat, lamb meat, ox tail, pork BBQ, smoked anchovies, smoked shrimp, stock fish bites, and date), respectively, using standard method agar (SMA; unless otherwise stated, all media were from Bacto, BD, Sparks, MD) for total aerobic mesophile counts after incubation at 36 °C for 48 h. For the packaging surfaces of hot dog and sausages, the detection limit was 1.3 log CFU/cm^3^. 

The level of total coliforms and *E. coli* (hygienic condition indicator) were determined [17] using the three-tube most-probable-number (MPN) evaluation with a detection limit of three cells per g for blendable samples (detection limit of 0.48 log CFU/g of samples) and for non-blendable samples (detection limit of 0.48 log CFU/mL of wash fluid). For the packaging surfaces of hot dog and sausages, the detection limit was 0.78 log CFU/cm^3^. After incubation at 36 °C for 48 h, a loopful of culture from each lauryl sulfate tryptose broth tube that produced gas was transferred to brilliant green bile broth (BGBB) and EC broth containing 4-methylumbelliferyl-b-D-glucuronide (EC-mug), respectively. After incubation for 48 h, BGBB tubes with growth and gas production at 36 °C confirmed the presence of coliforms, and EC-mug tubes with growth at 45.5 °C and fluorescence under long-wave UV light at 365 nm indicated the presence of *E. coli*. All positive EC tubes were streaked on eosine–methylene blue agar; purple colonies (with or without a green metallic sheen) were evaluated by API 20E test strips (bioMe’rieux, Hazelwood, MO, USA) for *E. coli* confirmation. One randomly selected and confirmed isolate from each positive EC tube and API 20E test strip was used for further study.

Due to their association with top pathogens causing foodborne illnesses resulting in death reported in the United States [10], the prevalence of three genus consisting of *Campylobacter*, *Listeria*, and *Salmonella* was also evaluated. *Listeria*, *Salmonella*, and *Campylobacter* were identified using AOAC-approved or performance-tested methods [18]. For *Listeria*, each sample homogenized with sterile PW was enriched in the University of Vermont Medium (UVM) Listeria enrichment broth at 30 °C for 48 h before one loopful of the enrichment broth was surface streaked onto Oxford Listeria (OL) agar for isolation. The colonies of presumptive *Listeria* on OL agar were identified to species level with the API Listeria kit. For *Salmonella*, each sample homogenized with sterile PW was pre-enriched in buffered peptone water (225 mL) at 36 °C for 24 h, enriched in Rappaport-Vassiliadis broth at 42 °C for 18 h, and post-enriched in M broth at 36 °C for 8 h. M broth cultures with a positive response in the immunoassay were surface streaked onto xylose lysine desoxycholate (XLD) agar for isolation. The colonies of presumptive *Salmonella* on XLD agar were confirmed with the API 20E test. For *Campylobacter*, each sample homogenized with sterile PW was enriched in modified Bolton broth (OXCM0983, Oxoid Ltd., Basingstoke, UK) supplemented with 5% laked horse blood (R54072, Thermo Fisher/Remel^TM^, Lenexa, KS, USA) and Bolton broth selective supplement (OXSR0183E, Oxoid Ltd.) at 42 °C for 48 h. Enrichment broth cultures were surface-streaked onto a modified *Campylobacter* blood-free selective agar (CM0739) with cefoperazone and amphotericin B (Antibiotic Supplements SR0155E, Oxoid Ltd.) and incubated microaerobically using the AnaeroPack System with Pack-MicroAero (Mitsubishi Gas Chemical, New York, NY, USA) at 42 °C for 48 h. Colonies with *Campylobacter*-like morphology on the plates and Gram-negative seagull-like cell morphology under a light microscope were presumed to be *Campylobacter* [18]. 

For the confirmation of *Campylobacter* [19], *Campylobacter* DNA was extracted from presumptive *Campylobacter* isolates in Bolton broth using a boiling method. Briefly, 2 mL of broth was centrifuged at 11,180× *g* for 4 min, and the supernatant was discarded. One ml of molecular-grade water was added to the bacterial pellets and re-suspended into the solution by vortexing. The suspension was centrifuged again for 4 min at 11,180× *g*, and the supernatant was discarded. In the final step, 300 µL of molecular-grade water was added to the pellet and re-suspended by vortexing. This was followed by heating the bacterial suspension at 100 °C for 20 min. The sample was centrifuged at full speed (23,831× *g*) for 4 min, and the supernatant containing the DNA was transferred into new tubes. The DNA concentration was measured using a NanoDrop 2000C UV-VIS spectrometer (Thermo Scientific, Waltham, MA, USA) and subsequently stored at −80 °C until PCR was performed.

A conventional PCR using primers targeting the *Campylobacter* spp. conserved 23sRNA gene was used to confirm isolates. Isolates were confirmed using a *C. jejuni*- and *C. fetus*-specific SYBR green-based real-time PCR assay (Catalog No. A25741, SYBR^®^ Green PCR Master Mix-Life Technologies, MA, USA). The forward and reverse primer sequence used for *C. jejuni* was TATACCGGTAAGGAGTGCTGGAG and ATCAATTAACCTTCGAGCACCG (a 650 bp region of conserved 23sRNA gene), respectively. The forward and reverse primer sequence used for *C. fetus* was GCAAATATAAATGTAAGCGGAGAG and TGCAGCGGCCCCACCTAT, respectively. The conventional PCR protocol utilized the Amplitaq 360 gold master mix kit with recommended conditions at an annealing temperature of 60 °C. Positive and negative controls were included for all reactions. Samples were run on a 1.5% agarose gel and visualized under UV light using the E-Gel Imager UV Light Base (Life Technologies Corp., Neve Yamin, Israel). 

All confirmed *Campylobacter*, *E. coli*, *Listeria*, and *Salmonella* isolates obtained above were suspended in Brucella broth containing 20% glycerol and stored at −80 °C until used for further evaluation of AMR.

### 2.4. Antimicrobial Resistance (AMR)

Following the procedure described by [20], antimicrobial susceptibility tests were performed on Mueller Hinton Agar (MHA) using the Kirby-Bauer disk diffusion method [21]. In brief, the confirmed *Campylobacter*, *E. coli*, *Listeria*, and *Salmonella* isolates were tested for susceptibility to 12 antimicrobial agents approved by the US Food and Drug Administration for clinical use, and their categories are shown in Table 1. The following 12 antimicrobial agents acquired from Oxoid, Ltd. were tested: ampicillin, amoxicillin-clavulonic acid, meropenem, amikacin, gentamycin, streptomycin, tobramycin, tetracycline, ciprofloxacin, nalidixic acid, chloramphenicol, and trimethoprim-sulfamethoxazole. Antimicrobial susceptibility, classified as “susceptible”, “intermediate”, and “resistant”, was interpreted in accordance with criteria established by the National Committee of Clinical Laboratory Standards [21]. In addition, bacteria classified as either resistant or intermediate were defined as “non-susceptible”, and those exhibiting resistance to at least one antimicrobial agent in three or more antimicrobial categories were defined as multi-drug resistant [22,23]. *E. coli* ATCC 25922 was used as a control strain for the performance of antimicrobials used in this study.

#### 2.4.1. Campylobacter

One loop of confirmed *Campylobacter* was transferred into 5 mL of modified Bolton broth (MBB) and incubated at 42 °C for 24 h. Then, 0.1 mL of each *Campylobacter* in MBB was transferred into a new 10 mL MBB and incubated at 42° for 24 h. One-tenth ml of each MBB, which was adjusted to approximately 8 log CFU/mL, was transferred to blood agar and spread uniformly. Before applying the antimicrobial discs, the plates were left for 10 min to allow any excess surface moisture to be absorbed. Then, antimicrobial discs were transferred using a 12-capacity disc dispenser. Plates were incubated for 24 h at 42 °C with a Pack-MicroAero, and the inhibition diameter zones were measured in millimeters for each antimicrobial agent with a caliper and recorded for each sample. *E. coli* ATCC 25922 cultured and sub-cultured in Mueller–Hinton Broth (MHB) as described under Section 2.4.2. *E. coli*, *Listeria*, and *Salmonella* sections were used as a control.

#### 2.4.2. *E. coli*, *Listeria*, and *Salmonella*

One loop of each confirmed *E. coli*, *Listeria*, and *Salmonella* isolate was transferred to 10 mL of MHB and incubated at 36 °C for 24 h. The isolates were again sub-cultured in MHB to ensure that they were all viable and fresh before antimicrobial resistance testing. *E. coli* ATCC 25922 was also cultured and sub-cultured similarly in MHB. One-tenth ml of each MHB, adjusted to approximately 8 log CFU/mL, was transferred to MHA plates and spread uniformly. Before applying the antimicrobial discs, the plates were left for 10 min to allow any excess surface moisture to be absorbed. Then, antimicrobial discs were transferred onto the plates using a 12-capacity disc dispenser. Plates were incubated for 24 h at 36 °C, and the inhibition diameter zones were measured in millimeters for the plate with a caliper and recorded for each sample. 

### 2.5. Data Analysis

Log-transformed microbial (aerobic mesophile, coliform, and *E. coli*) populations obtained from duplicates of each sample were averaged and subjected to an analysis of variance and Duncan’s multiple range test (SAS Institute, Cary, NC, USA) to determine the significance of the differences (*p* < 0.05) in mean populations of microorganisms. For the data below the quantification/detection level, values of detection level were used for statistical analysis purposes. SAS correlation analysis (SAS Institute, Cary, NC, USA) was implemented to evaluate prevalence relationship among *Campylobacter*, *E. coli*, *Listeria*, and *Salmonella* investigated (0 = absent; 1 = present). Associations were considered significant when *p* < 0.05.

## 3. Results and Discussion

### 3.1. Food Safety Inspection

Using the food safety inspection checklist (Table 2) adopted and simplified from the VBDH, MCDH, and SDDH checklists, 15 (approximately 79%) out of 19 markets assessed did not comply with at least one element in the checklist. More specifically, approximately 44% (4/9) of LCSMs and 90% (9/10) of SIOM outlets were out of compliance for at least one element. However, 50% of SIOMs had at least one item with no clear indication of an expiration date or sell-by date. In Packaging, 100% of LCSMs sell products in wholesome and good condition, while 20% of SIOMs had blood stains on the shelving in the meat storage section of refrigerators. In Storage, approximately 11% of LCSMs and 20% of SIOMs were observed with inappropriate separation of raw meat and poultry from produce in refrigerators. Shelving in refrigerators and freezers was maintained unclean in 20% of SIOMs, with pieces of plastics, Styrofoam, and gunk on the shelving rack wires. In addition, 20% of SIOMs did not comply with the guideline of the health department for the food storage area that should be clean and organized. However, there was no evidence of pests in both types of market. As for the Holding Temperature, only 77.8% of LCSMs and 50% of SIOMs maintained the refrigerator temperature at 41 °F (5 °C) or below. It was noted that a high percentage (50%) of violations at SIOMs was associated with the elements of missing or broken thermometers or refrigerator temperatures set above recommended range in the category of Holding Temperature, which may directly reflect the economic viability in the area. In addition, a rusty pipe was found with meat products in the refrigerator at one SIOM. All frozen products sold at LCSMs were maintained in a freezer temperature at or below 0 °F (−18 °C), while 20% of SIOMs had freezer temperatures above 0 °F.

In Employee Practices, approximately 22% of LCSMs and 30% of SIOMs employees did not follow health department guidelines for using good hygiene practices while handling food. The majority (60%) of violations at LCSMs were associated with the employee hygiene element of not wearing hair restraints or gloves, whereas 40% (4/10) of SIOM outlets were out of compliance in the category. In addition, employees in 10% of SIOMs were observed to consume food in food preparation areas. Using the food safety inspection checklist (Table 2), the rate of food safety compliance for LCSMs and SIOMs varied considerably, ranging from 77.8% (seven out of nine LCSMs) to 100% and from 50.0% (five out of ten SIOMs) to 100%, respectively. Overall, the average compliance rates of LCSMs and SIOMs were 94.5% and 76.0%, respectively.

### 3.2. Expiration Date/Use-by-Date and Price

Out of seven assessed food product types (Table 3), both LCSMs and SIOMs sold four products, including bacon, chicken leg, chicken wing, and sausage, at about the same price showing no significant difference (*p* > 0.05). However, prices of ground beef ($4.49 ± 0.62/lb) and pork chop ($4.25 ± 0.34/lb) at LCSMs were about 18.4% and 19.6% higher than the similar products ($3.61 ± 0.34/lb for ground beef and $3.47 ± 0.53/lb for pork chop) sold at SIOMs, whereas hot dog was about 32.3% lower at LCSMs ($1.28 ± 0.16/lb) compared to SIOMs ($1.92 ± 0.32/lb). Although no labels were available for specific ingredients used in ground beef and pork chop sold at SIOMs, further in-depth study on the differences of detailed composition (i.e., nutrition facts) in meat products and their relation to the sale price at both LCSMs and SIOMs would be interesting. In the meantime, the higher price of the same brand name commodity (hot dogs) sold at SIOMs than those sold at LCSMs may manifest an increased burden to economically challenged residents in food desert areas. Some of the SIOMs studied in the area obtain the product from LCSMs and market the product to consumers with a small price markup (personal communications). Caspi et al. [24] examined differences in major staple food pricing between small stores and supermarkets in Minnesota urban areas. They also found higher prices in smaller food stores than in supermarkets. Researchers [2,25,26] reported that supermarkets obtain food at wholesale prices and have more efficient economies of scale than smaller food stores, resulting in lower prices. However, several studies indicated that price variability for the same product does not always follow expected patterns [27,28,29,30,31], as our study demonstrated lower prices for ground beef and pork chop at SIOMs than at LCSMs. 

Additionally, in our study, the majority (71.4%, 5/7) of products sold at SIOMs had a shorter use-by date than those sold at LCSMs, with significance (*p* < 0.05) for bacon, sausage, and hot dog. Furthermore, 14 samples (at least two samples of all types) procured from SIOMs had no indication of either a use-by date or expiration date, and two sausages procured from SIOMs were 14 days past the use-by date. The findings reveal fewer quantities, less frequent turnover, and a shorter shelf life (due to being obtained from LCSMs) of products sold at SIOMs than those sold at LCSMs (personal communications).

Overall, the observed difference between LCSMs and SIOMs include: (1) a relatively limited variety and quantity of products were available at SIOMs, (2) a higher number of products sold at SIOMs had unknown use-by dates, (3) the sale price at both LCSMs and SIOMs was about the same except for a few commodities, (4) the turnover rate for ownerships in SIOMs was higher than those in LCSMs, and (5) the majority of owners of SIOMs consisted of diverse ethnicities.

### 3.3. Microbial Evaluation

#### 3.3.1. Meat Product (Section I) 

Results of the levels of aerobic mesophile, coliform, and *E. coli* counts in the 220 samples analyzed are shown in Table 4, Table 5 and Table 6. 

Among the blendable samples (Table 4), the mean aerobic mesophile counts were highest in ground beef (5.84 ± 1.26 log CFU/g) and lowest in bacon (3.34 ± 1.73 log CFU/g) for LCSMs and highest in pork sausage (7.06 ± 1.54 log CFU/g) and lowest in bacon (4.47 ± 1.99 log CFU/g) for SIOMs. Overall, the quality of meat samples based on the limits established by the International Commission on Microbiological Specifications for Foods (ICMSF 1986) indicated that samples acquired from SIOMs had higher levels of marginally acceptable (5 × 10^5^ to 5 × 10^7^ CFU/g) and unacceptable (≥5 × 10^7^ CFU/g) aerobic mesophiles. Among the non-blendable samples (Table 5), the mean aerobic mesophile counts were highest in pork chop (4.34 ± 0.87 log CFU/mL) and lowest in chicken leg (3.74 ± 1.99 log CFU/mL) for LCSMs and highest in pork chop (6.21 ± 1.19 log CFU/mL) and lowest in chicken wing (5.84 ± 2.34 log CFU/mL) for SIOMs.

Although there was limited availability of the same commodities at different stores in the studied food desert area, aerobic mesophile counts in the majority of sample types (beef cuts, pork sausage, chicken leg, chicken wing, and pork chop) obtained from SIOMs were significantly higher (*p* < 0.05) than those obtained from LCSMs (Table 4 and Table 5). The majority (75–78%) of beef cuts and pork sausages obtained from SIOMs had aerobic mesophile counts > 7 log CFU/g (Table 4). The aerobic mesophile counts in at least one out of all sample types except bacon obtained from SIOMs were even greater than 8.0 log CFU/g, consisting of 17% of the samples. Comparable to our findings, a study [6] conducted in Philadelphia, PA, also reported a higher level of aerobic mesophile counts in the majority of food products obtained from low-socioeconomic status (SES) markets compared with those found from high-SES markets. They indicated that populations of low SES might be more likely to experience food products of poorer microbial quality, which may impact both the product’s appeal and potential safety.

Among all seven commodities studied in the current study, on average, ground beef (2.03 ± 1.15 log MPN/g) and beef cuts (3.35 ± 1.86 log MPN/g) obtained from LCSMs and SIOMs had the highest coliform counts, respectively (data now shown). Coliform counts in all samples obtained from SIOMs were higher than those obtained from LCSMs, with significance (*p* < 0.05) in the bacon, beef cut, chicken leg, chicken wing, and pork chop samples. The highest coliform counts (5.34 log MPN/g or mL) were found in beef cut and chicken leg procured from SIOMs. This finding is concerning, since a high level of coliform counts generally indicates an unsanitary condition or poor hygiene practices during or after food production [33,34].

In addition, the average level of *E. coli* counts detected in beef cut (0.94 ± 0.65 log MPN/g) and pork sausage (1.04 ± 0.57 log MPN/g) procured from SIOMs and LCSMs, respectively, was the highest. Although no significant difference (*p* > 0.05) in *E. coli* counts was observed on all samples due to market source, the relatively high number of *E. coli* counts were observed in a ground beef (2.18 log MPN/g) and pork sausage (2.38 log MPN/g), both obtained from LCSMs (Table 4). This finding is concerning because their presence strongly indicates sewage and/or animal waste contamination [35,36]. 

Regarding the higher microbial levels observed in SIOM samples than LCSM-acquired ones in the present study, we speculate that the food products in SIOMs may have been exposed to variations in temperature during display or storage in the refrigerator. In contrast, LCSM products, on average, were well-maintained at recommended temperatures (41 °F or below) or handled better (Table 2, overall compliance rate of 94.5% for LCSMs vs. 76.0% for SIOMs). In the evaluation of fresh produce samples, Kim et al. [37] reported compatible findings that the majority of samples acquired from SIOMs had higher counts of aerobic mesophiles, coliforms, and *E. coli* than those obtained from LCSMs. In their study, SIOMs also had a lower compliance rate (85.4%) in the food safety inspection checklist than LCSMs (93.8%).

In the meantime, out of 220 samples collected during the study period, *E. coli*, *Campylobacter*, *Listeria* spp., and *Salmonella* were detected in 46 (20.9%), 8 (3.6%), 12 (5.5%), and 6 (2.7%), respectively, (Table 6). The overall prevalence of *E. coli* was detected in 16.7% (19/114) and 25.5% (27/106) of meat products obtained from LCSMs and SIOMs, respectively. Interestingly, a high prevalence (50%) of *E. coli* was detected in pork sausages obtained from both LCSMs and SIOMs. In addition, although 50% of tested samples were marginally acceptable in the level of aerobic mesophiles (Table 4), out of all samples, only beef cuts obtained from LCSMs resulted in the absence of *E. coli*, *Campylobacter*, *Listeria* spp., and *Salmonella *(Table 6). *E. coli* was detected in all sample types except bacon and beef cut obtained from LCSMs. Comparable to this finding, a study conducted by [38] also reported a high prevalence of *E. coli* in the ground beef (69%) and pork (44%) samples obtained from supermarkets in several U.S. states, indicating the dominance of unhygienic condition of meat products in the United States. All SIOM-acquired sample types except chicken wings had equal to or higher *E. coli* occurrence than LCSM-acquired samples. These findings are also consistent with the overall lower compliance rate at SIOMs than at LCSMs in the food safety inspection checklist (Table 2). Because the presence of *E. coli* in food products indicates possible fecal contamination during handling and/or processing, these findings should be taken seriously, given that pathogenic *E. coli* O157:H7 can often come from the same contamination sources [39,40]. 

The occurrence of *Campylobacter* was 1.8% (2/114) and 5.7% (6/106) of meat products obtained from LCSMs and SIOMs, respectively (Table 6). The majority (75%, 6/8) of *Campylobacter* detected in the samples were associated with SIOMs. SIOM-acquired sample types, except pork sausages, had higher *Campylobacter* occurrence than LCSM-acquired sample types. The occurrence of *Listeria* was 5.7% (6/106) and 5.3% (6/114) of meat products obtained from SIOMs and LCSMs, respectively. The rate of *Listeria* occurrence was about the same for the tested meat products obtained from both SIOMs and LCSMs. It is noteworthy that the presence of *Listeria* was associated with ground beef, chicken wing, and pork sausage obtained from SIOMs and bacon, chicken leg, pork chop, and pork sausages obtained from LCSMs. *Listeria* spp. detected from LCSM-acquired samples were *L. monocytogenes* and *L. welshimeri*, whereas *Listeria* spp. detected from SIOM-acquired samples were *Listeria monocytogenes*, *L. innocua*, and *L. welshimeri*. 

The majority (83.3%, 5/6) of *Salmonella* detected was from SIOM-acquired samples. Furthermore, the occurrence of *Salmonella* was mostly associated with pork sausages (66.7%, 4/6), regardless of market source. Although due to the limited availability of the same commodities at SIOMs and LCSMs, each commodity acquired in duplicate may not be representative of all the commodities in the tested food desert areas, chicken wings (18.8% *E. coli*, 12.5% *Campylobacter*, 18.8% *Listeria*, and 6.3% *Salmonella*) obtained from SIOMs and pork sausages (50% *E. coli*, 4.5% *Campylobacter*, 9% *Listeria*, and 4.5% *Salmonella*) obtained from LCSMs had the presence of multiple bacteria, resulting in a higher risk of foodborne illnesses. In addition, chicken wing, chicken leg, and ground beef obtained from one SIOM had multiple occurrences (either *E. coli*, *Campylobacter*, and *Listeria* or *E. coli*, *Campylobacter*, and *Salmonella*) of bacteria, indicating a potential lack of good handling practices in the store.

The prevalence of bacteria obtained in the current study was comparable to the prevalence found in our previous studies [20,37] that surveyed 122 fresh produce samples obtained from food desert markets and 194 value-added commodities, including meat and sausage products from farmers’ markets, respectively, in the same region of VA. In the farmers’ market study, the occurrence of *Campylobacter*, *E. coli*, *Listeria*, and *Salmonella* was 0.5%, 24.5%, 16.7%, and 1.0% of samples, respectively, while in the food desert study, the occurrence of *Campylobacter*, *E. coli*, and *Listeria* was 10.7%, 4.9%, and 3.3%, respectively. 

The overall prevalence of *E. coil* (20.9%), *Listeria* spp. (5.5%), and *Salmonella* (2.7%) detected in the meat products obtained from the current study was considerably higher than those [*E. coil* (4.9%), *Listeria* spp. (3.3%), and *Salmonella* (0%)] detected in fresh produce obtained from the studied food deserts. However, interestingly, the overall rate of *Campylobacter* (3.6%), known as being prevalent in food animals [41], that was detected in the present study was considerably lower than the rate detected in the fresh produce study (10.7%), indicating a potential occurrence of cross-contamination during or at the venue from farm to store. In addition, the overall occurrence of *Campylobacter* (0.5%) and *Salmonella* (1.0%) detected in the value-added products obtained from farmers’ markets in the same region was considerably lower than food desert-acquired meat products. However, the occurrence of *E. coli* (24.5%) and *Listeria* spp. (16.7%) in the value-added products was much higher than that of the food desert-acquired meat products. Differences among sample commodities (fresh produce vs. meat products vs. value-added products), mode of display (i.e., refrigeration and open-air) and transportation associated with farmers’ market, SIOMs, and LCSMs could have led to the disparity in results. Market sanitary conditions during production, processing, and retail, could also have contributed to the differences. Therefore, more information on contamination at different points in the production and supply chain associated with market sources is needed to interpret these differences. 

Although samples tested revealed the presence of *Campylobacter*, *E. coli*, *Listeria*, and *Salmonella*, overall, no correlation (r < 0.2266, *p* > 0.0006) among the prevalence of the bacteria was observed. In specific, Pearson correlation coefficients for the prevalence of *Campylobacter*, *E. coli*, *Listeria*, and *Salmonella* in meat products procured from SIOMs and LCSMs were r < 0.3326 with *p* > 0.0003 and r < 0.1867 with *p* > 0.0487, respectively. The very low correlation observed in the current study affirms our previous findings [20] that commodities with the presence of *E. coli*, the best bacterial indicator of fecal contamination [42], do not warrant the presence of harmful, disease-causing microorganisms or vice versa.

#### 3.3.2. Packaging Surface (Section II)

Further analysis was conducted to assess any difference in microbial quality on the packaging surfaces utilizing the same brand name hot dogs and sausages available at both SIOMs and LCSMs as a strategy to validate any difference in GHP between SIOMs and LCSMs. Results of the levels of aerobic mesophiles and coliforms, counts in the 61 samples analyzed are shown in Table 7. 

The mean aerobic mesophile counts (3.07 ± 1.62 log CFU/cm^3^) on the packaging surface of sausages obtained from SIOMs were the highest among the packaging surfaces tested. The levels on the packaging surfaces of sausages obtained from SIOMs were significantly higher (*p* < 0.05) than those (2.07 ± 1.14 log CFU/cm^3^) obtained from LCSMs. One of the sausage packaging surfaces obtained from SIOMs had an aerobic mesophile count of 7.31 log CFU/cm^3^, which may indicate cross-contamination of the packaging surface from blood-stained shelving and/or a content leakage through the packaging. Although one of the sausage packaging surfaces had the highest coliform count (2.68 log MPN/cm^3^), there was no significant difference (*p* > 0.05) in coliform counts on packaging surfaces, regardless of sample type and store source. In addition, *Campylobacter* was detected in 1 (6.7%) of 15 sausage packaging surfaces obtained from SIOMs. However, on none of the packaging surfaces analyzed in the current study were *E. coli*, *Listeria*, and *Salmonella* detected.

#### 3.3.3. Ethnic Product (Section III)

Due to the prevalence of ethnic food products in the studied SIOMs, which all are operated by foreign nationals, an assessment of microbial quality on the representative ethnic food products procured from eight SIOMs were conducted. The results of bacterial occurrence in the 45 samples analyzed are shown in Table 8 and Table 9. Although there was limited availability of the same commodities at different stores and even within each store in the studied food desert area, among blendable samples, dried okazi leaf had significantly higher (*p* < 0.05) counts of aerobic mesophiles (8.73 ± 1.8 log CFU/g), coliforms (5.04 ± 0.00 log MPN/g), and *E. coli* (1.43 ± 1.34 log MPN/g) than other products (Table 8). Among non-blendable samples, whole chicken had significantly higher (*p* < 0.05) counts of aerobic mesophiles (8.05 ± 0.02 log CFU/mL of washed fluid), coliforms (4.22 ± 1.05 log MPN/mL of wash fluid), and *E. coli* (0.86±0.00 log MPN/mL of wash fluid) than other products (Table 9). While the majority of non-blendable (73.9%, 17/23) samples tested had aerobic mesophile counts less than 5 log CFU/mL of wash fluid, 53.6% (15/28) of blendable samples had aerobic mesophile counts greater than 7 log CFU/g of samples). In addition, samples (i.e., whole chicken, seasoned meat, smoked anchovies, and saffron powder) labeled with an emphasis on ethnic origin exhibited aerobic mesophile counts of more than 7.45 log CFU/g. It is also concerning, given the consideration that some products such as seafood salad, smoked shrimp, and smoked anchovies may be consumed as ready-to-eat foods. Furthermore, *Campylobacter* and *Salmonella* were detected in both dried okazi leaf samples, while *E. coli* was also detected in one of the samples. *Listeria welshimeri* was detected on the packaging surface of one ground egusi sample.

### 3.4. Antimicrobial Resistance (AMR)

#### 3.4.1. *Campylobacter*

Prevalence of antimicrobial resistance in *Campylobacter*, *E. coli*, *Listeria*, and *Salmonella* isolates to the 12 antimicrobials tested are summarized in Table 10. The majority (71.4%) of the *Campylobacter* isolates obtained from our study showed MDR. The susceptible, intermediate, and resistant patterns of *Campylobacter* isolates obtained from LCSMs and SIOMs in relation to the antimicrobials tested are presented in Figure 1A,B, respectively. The two *Campylobacter* isolates detected in chicken leg and pork sausage obtained from LCSMs showed MDR (Figure 1A). These isolates were resistant to AMP, GEN, TOB, TCY, CHL, and SXT in 100% and to STR in 50%. In contrast, the isolates were susceptible to MEM and AMK. Both isolates displayed non-susceptibility to at least nine antimicrobials (Table 10). 

Of the total 14 *Campylobacter* isolates obtained, resistance to AMP and TOB was the most common (100%), followed by STR (85.7%), GEN (64.3%), AMC (50.0%), TCY (42.9%), SXT (42.9%), and CHL (35.7%) (Figure 1A,B).

Among the 12 *Campylobacter* isolates obtained from SIOMs, resistance to STR was the second-most common in 91.7%, followed by AMC and GEN in 58.3%, TCY and SXT in 33.3%, and CHL in 25% (Figure 1B). None of the isolates were fully susceptible to all antimicrobials tested in the current study. Approximately 66.7% (8/12) of the isolates detected in samples of bacon, chicken leg, chicken wing, ground beef packaging, and dried okazi leaf showed MDR. Furthermore, three isolates obtained from chicken leg, chicken wing, and dried okazi leaf displayed resistance to seven antimicrobials while one isolate obtained from dried okazi leaf displayed non-susceptibility to even 11 antimicrobials. Our previous studies [20,43] on the prevalence of AMR in *Campylobacter* isolates obtained from farmers’ markets, farm animals, wildlife, and food samples in the eastern United States found a similar pattern of non-susceptibility to all the antimicrobials tested. In detail, Kim et al. [43] reported that 97.4% of *Campylobacter* isolates obtained from farms were non-susceptible to at least one antimicrobial. Studies [20] also found that all *Campylobacter* isolates detected in value-added products, including sausages procured from farmers’ markets and in fresh produce procured from local stores in Central Virginia, were non-susceptible to at least one antimicrobial agent, and 91.7% and 84.6% of the isolates obtained from farmers’ markets and local stores, respectively, displayed MDR. Their resistance to AMP was the most common in 100%, followed by CHL (69.2–100%), SXT (69.2–100%), STR (53.8–100%), and AMC (53.8–100%).

Although there was a limited occurrence of *Campylobacter* isolates in the studied food desert area, the prevalence of MDR in the isolates obtained from LCSMs (100%) was much higher than the prevalence from SIOMs (66.7%). Lower prevalence of MDR and non-susceptibility (24.9% for isolates from SIOMs vs. 100% for isolates from LCSMs against more than nine antimicrobials) in *Campylobacter* isolates detected in samples obtained from SIOMs (Table 9) may be due to the economic viability of the small-scale production system, which mainly relies on organic processing. In the meantime, due to the management differences, *Campylobacter* isolates in samples obtained from LCSMs may have likely been exposed to the practice of antimicrobial usage during large-scale agricultural production systems. Several scientists [44,45] reported that the practice of antimicrobial usage in agricultural production influences the prevalence of AMR in bacteria. More importantly, although isolates obtained from LCSMs were susceptible to MEM and AMK, all isolates obtained from both LCSMs and SIOMs were resistant to at least three antimicrobial agents.

#### 3.4.2. *E. coli*

Among the 93 *E. coli* isolates, resistance to TCY was the most common in 65 (69.9%) isolates, followed by AMP (54.8%), AMC (28.0%), STR (19.4%), and SXT (14.0%) (Figure 2A,B). Thirty (32.3%) *E. coli* isolates showed MDR (Table 10). Only 16 (17.2%) isolates were susceptible to all tested antimicrobials, indicating that 82.8% of *E. coli* isolates were non-susceptible to at least one antimicrobial agent (Table 10). 

Among the 33 *E. coli* isolates obtained from LCSMs, resistance to TCY was the most common in 22 (66.7%) isolates, followed by AMP (51.5%), AMC (21.2%), STR (9.1%), and NAL (9.1%) (Figure 2A). Nine *E. coli* isolates detected in the majority of sausages (88.9%, 8/9) and ground beef obtained from LCSMs displayed MDR (Table 10). These isolates were resistant to AMP and TCY in 100% and to AMC in 55.6%. In contrast, the isolates were susceptible to MEM, AMK, TOB, and CIP. One isolate detected in ground beef displayed resistance and non-susceptibility to six and seven antimicrobials, respectively.

Out of the 60 *E. coli* isolates obtained from SIOMs, as illustrated above, resistance to TCY was the most common in 43 (71.7%) isolates, followed by AMP (56.7%), AMC (31.7%), STR (25.0%), SXT (20.0%), and NAL (10.0%) (Figure 2B). Twenty-one (35.0%) *E. coli* isolates obtained from sausages, chicken leg, chicken wing, meat seasoned, pork chop, ground beef, dried okazi leaf, and whole chicken displayed MDR (Table 10). The majority (52.4%, 11/21) of MDR was associated with sausages. These MDR isolates were resistant to AMP in 100% and non-susceptible to AMC and TCY in 100%. In contrast, 100% of the isolates were susceptible to MEM, AMK, TOB, and CIP. Each isolate detected in both whole chickens with ethnic origin labels obtained from SIOMs showed resistance to seven antimicrobials. The AMR prevalence of *E. coli* found in the current study was comparable to that in our previous studies [20,43]. In the study on *E. coli* detected in farm animals, wildlife, and food samples in the Eastern United States [43], resistance to TCY was the most common in 62.1% (41/66), followed by AMP (50%), TOB (16.7%), and STR (10.6%) while approximately 20% of them displayed MDR. In the study on value-added products [20], including sausages procured from farmers’ markets, resistance to TCY was again the most common in 35 (29.7%) isolates, followed by AMP (28.8%), STR (22.0%), and AMC (14.4%). Approximately 16% of the *E. coli* detected in ground beef, ground pork, lamb loin chop, and pork sausage revealed MDR. Additionally, *E. coli* detected in fresh produce procured from local stores in the same region of Central Virginia demonstrated the most resistance to AMP, NAL, CHL, and SXT in 27.3%, followed by AMC (18.2%), MEM (18.2%), STR (18.2%), and TCY (18.2%). Approximately 27% of them displayed MDR.

Overall, the current study revealed a lower prevalence of MDR in the isolates obtained from LCSMs (27.3%) than those from SIOMs (35%). However, the prevalence of non-susceptibility in *E. coli* isolated from LCSMs and SIOMs was about the same, displaying approximately 85% and 82%, respectively, to more than one antimicrobial agent. In addition, one and two isolates from LCSMs and SIOMs, respectively, displayed non-susceptibility to seven antimicrobials. Since the presence of *E. coli* is mostly associated with good handling practices (i.e., hygiene practices) rather than agricultural practices (i.e., application of antimicrobials), a slightly higher prevalence of MDR and lower prevalence of non-susceptibility in *E. coli* isolates obtained from SIOMs samples may not necessarily represent the exposure of the bacteria to antimicrobials during the production and processing practices.

#### 3.4.3. *Listeria*

Among the 15 *Listeria* isolates detected, resistance to NAL was the most common in 100%, followed by STR (20%), AMK (13.3%), TCY (13.3%), and MEM (6.7%) (Figure 3A,B). None of the *Listeria* tested was susceptible to all tested antimicrobials, indicating that 100% of *Listeria* isolates obtained were resistant to at least one antimicrobial agent (Table 10). 

In detail, out of nine *Listeria* isolates obtained from LCSMs, resistance to MEM, AMK, and STR was the second most common in each (11.1%) isolate (Figure 3A). Although one isolate detected in sausage displayed resistance and non-susceptibility to two antimicrobials in two categories and to six antimicrobials, respectively, none of the isolates displayed MDR (Table 10). This isolate was resistant to AMP, STR, and NAL. In contrast, the isolate was susceptible to AMC, AMK, GEN, CIP, CHL, and SXT. For *Listeria* isolates obtained from SIOMs (Figure 3B), resistance to STR and TCY was the second most common in two (33.3%) isolates, followed by AMK (16.7%). One isolate detected in ground beef and chicken wing displayed resistance to three and four antimicrobials, respectively, in three categories meeting MDR classification. In addition, the isolate obtained from the ground beef showed non-susceptibility to five antimicrobials. In the study [20] on value-added products, including sausages procured from farmers’ markets, they also found that the resistance of *Listeria* isolates to NAL was the most common in 79.4%, followed by MEM (44.1%) and AMP (38.2%). Approximately 59% of the *Listeria* isolates obtained from ground beef, sausage, beef tail, and lamb bone were MDR. In contrast, the most effective antimicrobial tested in their study was TCY showing 64.7% susceptibility.

Overall, although the AMR in *Listeria* isolates obtained from both market sources in the current study demonstrated similar patterns, the prevalence of MDR in the isolates obtained from SIOMs (33.3%) was much higher than the prevalence from LCSMs (0%). However, all isolates obtained from both market sources displayed resistance and non-susceptibility to at least one antimicrobial agent. Furthermore, non-susceptibility to more than five antimicrobials was much higher for LCSM-acquired isolates (22.2%) than SIOM-acquired isolates (16.7%). Therefore, as addressed earlier, since the presence of *E. coli* and *Listeria* is mostly associated with environmental sanitary conditions [33,46,47,48] and handling practices (i.e., hygiene practices), rather than agricultural practices (i.e., application of antimicrobials), a higher prevalence of MDR alone in *Listeria* isolates obtained from SIOMs samples may not necessarily represent the exposure of the bacteria to antimicrobials during the production and processing practices. Therefore, continued research on a larger-scale sample size with a greater diversity of products is warranted to determine the cause(s) of the observed and somewhat contradictory differences to *Campylobacter* in the prevalence of the pathogens and AMR profiles between SIOMs and LCSMs.

#### 3.4.4. *Salmonella*

Among the eight *Salmonella* isolates, resistance to TCY was the most common in 50%, followed by AMP (37.5%), NAL (37.5%), and AMC (12.5%) (Figure 4A,B). One *Salmonella* isolate detected in a sausage obtained from LCSMs displayed resistance to AMP and TCY (Figure 4A). Out of seven isolates detected in sausages, chicken leg, dried okazi leaf, and chicken wing obtained from SIOMs, only two isolates detected in sausages were susceptible to all antimicrobials tested. The other five isolates displayed resistance to at least one antimicrobial agent. For isolates obtained from SIOMs (Figure 4B), resistance to TCY and NAL was the most common in three (42.9%) isolates, followed by AMP (28.6%) and AMC (14.3%). One isolate detected in a sausage isolated from SIOMs showed MDR (Table 10). Although a limited number (two isolates) of *Salmonella* was detected in our previous study [20] on value-added commodities, the AMR prevalence of *Salmonella* found in the study revealed that resistance to AMP, AMC, STR, and TCY was the most common in 100% and both isolates displayed MDR. In addition, another study [43] on the prevalence of AMR in 121 *Salmonella* isolates obtained from farm animals, wildlife, and food samples also demonstrated that resistance to TCY was most common (16 isolates, 13.2%), followed by resistance to STR (12 isolates, 9.9%), and AMP (10 isolates, 8.3%). Approximately 93% (112/121) of the bacteria were non-susceptible to at least one of the 12 antimicrobials. Based on the findings from studies [43,49,50,51,52,53,54], as indicated by Kim et al. [20], MEM and AMK may be the most effective antimicrobials in the treatment of *Salmonella* infections in veterinary and human medical practices. 

Although the presence of *Salmonella* may be associated with agricultural practices, due to a limited number of *Salmonella* assessed, a lower prevalence of MDR and non-susceptibility to antimicrobials found in one *Salmonella* isolate obtained from LCSMs samples may not necessarily represent any potential impact of production and processing practices on the development of bacterial prevalence in AMR. Therefore, continued research efforts on a larger-scale sample size with a greater diversity of products are warranted to determine the cause(s) of the observed and somewhat contradictory differences in the prevalence of the pathogens and AMR profiles between SIOMs and LCSMs.

Overall, the present survey revealed that the prevalence of MDR to 12 antimicrobials tested was the highest in *Campylobacter* (71.4%), followed by *E. coli* (32.3%), *Listeria* spp. (13.3%), and *Salmonella* (12.5%) (Table 10). Among all the tested antimicrobials, AMP (100%) and TOB (100%) showed the highest frequency of resistance among *Campylobacter*, TCY (69.9%) among *E. coli*, NAL (100%) among *Listeria*, and TCY (50%) among *Salmonella*, respectively, demonstrating different resistance patterns among the bacteria in this study (Figure 1, Figure 2, Figure 3 and Figure 4). The most effective antimicrobials tested in this study were CIP and NAL for *Campylobacter*, AMK, MEM, and TOB for *E. coli*, AMC, AMP, CHL, CIP, GEN, SXT, and TOB for *Listeria*, and AMK, CHL, CIP, GEN, MEM, STR, SXT, and TOB for *Salmonella*. The most effective antimicrobial was CIP showing 100% susceptibility to *Campylobacter*, *Listeria*, and *Salmonella* and TOB to *E. coli*, *Listeria*, and *Salmonella*. However, none of the bacterial species assessed in the current study showed 100% susceptibility to all antimicrobials. Findings from the present study revealed diverse AMR profiles and specificity regarding the food product type, market source, and bacterial species in the studied food desert area.

## 4. Conclusions

Findings demonstrated a potential for bacteriological health hazards associated with food products obtained from LCSMs and SIOMs in the studied food desert area. Additionally, it emphasizes the importance of and need for good handling practices regardless of the market source. The food safety compliance rate of LCSMs following the health department guidelines was higher compared to that of SIOMs, as demonstrated by the lower levels of aerobic mesophile counts and coliforms in all the tested samples, and lower prevalence of *E. coli*, *Campylobacter*, and *Salmonella* in the majority of samples. The microbial quality study on the packaging surfaces of samples procured from LCSMs and SIOMs validated the lack of GHP in SIOMs. The lower prevalence of MDR and non-susceptibility in *Campylobacter* isolates and the lower prevalence of non-susceptibility in *Listeria* isolates detected in SIOM-acquired food products than those isolates detected in LCSM-acquired samples indicate that the difference of AMR prevalence in bacteria may have to do with the practices of small-scale producers. The higher price of the same brand name commodity sold at SIOMs than those sold at LCSMs may manifest an increased burden to economically challenged residents in food desert areas. Furthermore, the majority of products sold at SIOMs had a shorter use-by-date than those sold at LCSMs. These findings may also indicate increased food safety risks associated with economic viability in food desert areas. Therefore, elaborated and in-depth research on a larger-scale sample size with a greater diversity of products is needed to determine and intervene the cause(s) of the observed differences in the prevalence of the pathogens and AMR profiles. Concerted research and extension efforts with proper food safety education and training will assist food outlet vendors and managers in at-risk, low-income communities to better comply with food safety guidelines and regulations, in order to reduce the risk of foodborne illness for their consumers.

## Figures and Tables

**Figure 1 pathogens-12-00965-f001:**
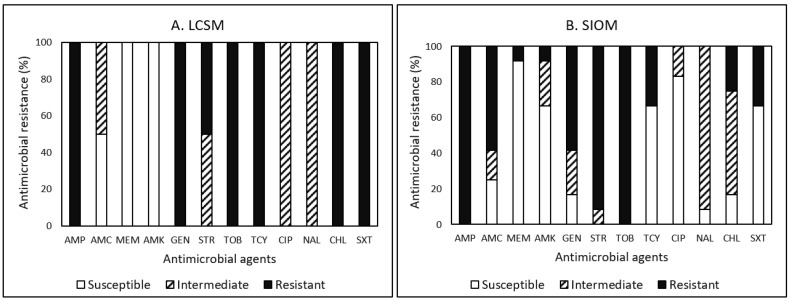
(**A**,**B**) Prevalence of resistance to 12 antimicrobial agents in 2 and 12 *Campylobacter* isolates in food products procured from large chain supermarkets (LCSM) and small, independently owned corner markets and convenience stores (SIOM), respectively.

**Figure 2 pathogens-12-00965-f002:**
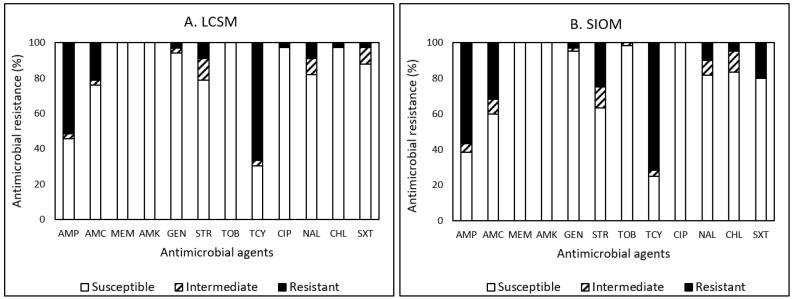
(**A**,**B**) Prevalence of resistance to 12 antimicrobial agents in 33 and 60 *E. coli* isolates in food products procured from large chain super markets (LCSMs) and small, independently owned corner markets and convenience stores (SIOMs), respectively.

**Figure 3 pathogens-12-00965-f003:**
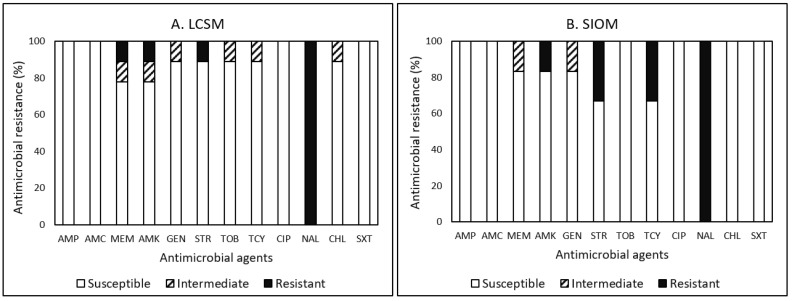
(**A**,**B**) Prevalence of resistance to 12 antimicrobial agents in 9 and 6 *Listeria* isolates in food products procured from large chain super markets (LCSMs) and small, independently owned corner markets and convenience stores (SIOMs).

**Figure 4 pathogens-12-00965-f004:**
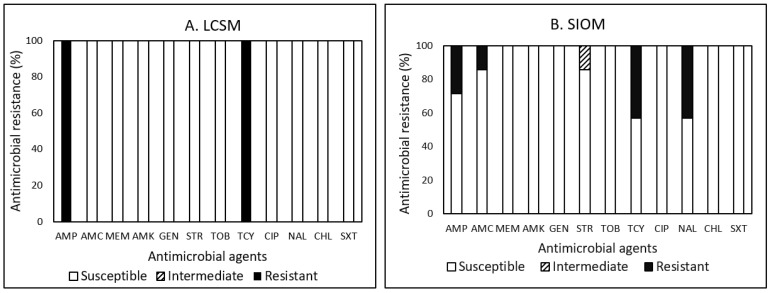
(**A**,**B**). Prevalence of resistance to 12 antimicrobial agents in 1 and 7 *Salmonella* isolates in food products procured from large chain supermarkets (LCSMs) and small, independently owned corner markets and convenience stores (SIOMs).

**Table 1 pathogens-12-00965-t001:** A list of antimicrobials and interpretive criteria used in this study (CLSI 2015) *.

Antimicrobial Category	Antimicrobial Agent and Its Abbreviation	Concentration (µg/disk)	Zone Diameter (mm)		
			S	I	R
Penicillins	Ampicillin (AMP)	10	>17	14–16	<13
β-lactamase inhibitor combinations	Amoxicillin-clavulanic acid (AMC)	30	>18	14–17	<13
Carbapenems	Meropenem (MEM)	10	>23	20–22	<19
Aminoglycosides	Amikacin (AMK)	30	>17	15–16	<14
	Gentamicin (GEN)	10	>15	13–14	<12
	Streptomycin (STR)	10	>15	12–14	<11
	Tobramycin (TOB)	10	>15	13–14	<12
Tetracyclines	Tetracycline (TCY)	30	>15	12–14	<11
Fluoroquinolones	Ciprofloxacin (CIP)	5	>21	16–20	<15
Quinolones	Nalidixic acid (NAL)	30	>19	14–18	<13
Phenicols	Chloramphenicol (CHL)	30	>18	13–17	<12
Folate pathway inhibitors	Trimethoprim-sulfamethoxazole (SXT)	25	>16	11–15	<10

* Interpretive criteria: S, susceptible; I, intermediate; and R, resistant to antimicrobial agents tested.

**Table 2 pathogens-12-00965-t002:** Checklist of food safety inspections conducted and market compliance *.

Food Safety Checklist	Food Outlets/Compliance (%)	
	LCSMs (*n* = 9)	SIOMs (*n* = 10)
**Packaging**		
Clear indication of expiration date (or sell by)	100	50.0
Good condition (e.g., no damage and no evidence of fluid leakage)	100	80.0
**Storage**		
Raw meat & poultry stored separate or below produce	88.9	80.0
No evidence of pest is present	100	80.0
Dry product storage area are clean and organized	100	80.0
Refrigerators and freezers maintained clean (shelving, etc.)	100	80.0
**Holding Temperature**		
Product is stored or displayed at 41 °F (5 °C) or below	77.8	50.0
Frozen foods are frozen in freezer temperature at or below 0 °F (−18 °C)	100	80.0
**Employee Practices**		
Employees use good hygiene practices wearing a hair net and gloves while handling products	77.8	70.0
Employees do not consume food in product storage and preparation areas	100	90.0
**Overall**	94.5	76.0

* The checklist was excerpted from the Virginia Beach Department of Public Health [14], Illinois McHenry County Department of Health [15], and South Dakota Department of Health [16] educational materials and adapted for this study; LCSMs: large chain supermarkets; SIOMs: small, independently owned corner markets and convenience stores; n: number of food outlets assessed.

**Table 3 pathogens-12-00965-t003:** Summary of products assessed for the comparison of expiration date or use-by-date and price, and their market sources in food desert areas of Virginia studied from August 2018 to March 2020 *.

Sample Type	Market Source (Store Number)					
	LCSMs (*n* = 9)			SIOMs (*n* = 10)		
	Sample Number	Price ($/lb)	Use-by Date (Day)	Sample Number	Price ($/lb)	Use-by Date (Day)
Bacon	16	4.36 ± 1.98 a	82.8 ± 17.7 A	10	5.57 ± 1.31 a	19.3 ± 7.9 B
Ground beef	20	4.49 ± 0.62 a	3.2 ± 2.2 A	22	3.61 ± 0.34 b	4.0 ± 1.3 A
Chicken leg	18	1.51 ± 0.49 a	3.8 ± 1.0 A	16	1.19 ± 0.36 a	5.0 ± 1.2 A
Chicken wing	16	2.96 ± 1.16 a	4.5 ± 1.6 A	16	3.21 ± 0.46 a	4.3 ± 1.6 A
Pork chop	18	4.25 ± 0.34 a	4.0 ± 2.0 A	16	3.47 ± 0.53 b	3.3 ± 2.1 A
Sausage	22	1.44 ± 0.64 a	76.0 ± 37.2 A	15	2.72 ± 2.08 a	19.0 ± 21.2 B ^#^
Hot dog	12	1.28 ± 0.16 b	109.2 ± 23.7 A	12	1.92 ± 0.32 a	72.7 ± 43.1 B

* LCSMs: large chain supermarkets and SIOMs: small, independently owned corner markets and convenience stores; “Use-by-date” or “Expiration date” is defined as “Use-by date” for illustration purposes and is calculated either back (days before use-by date) or forward (days past use-by date) from the date when the sample was procured from the store; ^#^ two samples procured from SIOMs had 14 days past use-by date; a total of 281 samples consisted of 148 and 133 samples from 9 LCSMs and 10 SIOMs, respectively; samples were purchased in duplicate from each of market source; the price excludes tax; in the same row within the same sample type, means followed by the same uppercase letter and lowercase letter, respectively, are not significantly different (*p* > 0.05).

**Table 4 pathogens-12-00965-t004:** Level of microorganisms in the blendable samples procured from large chain supermarkets (LCSMs) and small, independently owned corner markets and convenience stores (SIOMs) *.

Sample Type	Sample Number	Store Source	Microorganism Population (log CFU/g or log MPN/g)			
			Aerobic Mesophiles		Coliforms	*E. coli*
			Average	ICMSF (%) ^#^	Range	Range
Bacon	16	LCSMs	A 3.33 ± 1.73 b	88, 0, 12	0.48–0.56	0.48–0.48
	10	SIOMs	A 4.47 ± 1.99 b	70, 10, 20	0.48–1.97	0.48–0.96
Beef cut	4	LCSMs	B 3.98 ± 2.3 b	50, 50, 0	0.48–0.48	0.48–0.48
	8	SIOMs	A 6.99 ± 1.86 a	25, 0, 75	0.48–5.34	0.48–1.97
Ground beef	20	LCSMs	A 5.85 ± 1.26 a	35, 30, 35	0.48–4.18	0.48–2.18
	22	SIOMs	A 6.53 ± 1.56 a	18, 18, 64	0.48–5.04	0.48–1.58
Pork sausage	22	LCSMs	B 4.75 ± 1.33 ab	73, 23, 4	0.48–4.38	0.48–2.38
	18	SIOMs	A 7.06 ± 1.54 a	22, 0, 78	0.48–3.38	0.48–1.63

* In the level of aerobic mesophiles within the same sample type, means preceded by the same uppercase letter are not significantly different (*p* > 0.05); in the same column within the same store source, means followed by the same lowercase letter are not significantly different (*p* > 0.05); samples are obtained from 9 LCSMs and 10 SIOMs, respectively; *^#^* values are the percentages of samples with aerobic mesophile counts within the recommended range for good quality (≤5 × 10^5^ CFU/g), marginally acceptable (5 × 10^5^ to 5 × 10^7^ CFU/g), and unacceptable (≥5 × 10^7^ CFU/g), respectively, according to the limits established by the International Commission on Microbiological Specifications for Foods [32].

**Table 5 pathogens-12-00965-t005:** Level of microorganisms in the non-blendable samples procured from large chain supermarkets (LCSMs) and small, independently owned corner markets and convenience stores (SIOMs) *.

Sample Type	Sample Number	Store Source	Microorganism Population (log CFU/mL of Wash Fluid or log MPN/mL of Wash Fluid)			
			Aerobic Mesophiles		Coliforms	*E. coli*
			Average	Range	Range	Range
Chicken leg	18	LCSMs	B 3.74 ± 1.99 a	1.30–7.53	0.78–2.75	0.78–0.86
	16	SIOMs	A 5.90 ± 2.11 a	1.60–8.56	0.78–5.34	0.78–1.93
Chicken wing	16	LCSMs	B 4.30 ± 1.76 a	2.00–8.24	0.78–3.34	0.78–1.66
	16	SIOMs	A 5.84 ± 2.34 a	3.88–7.52	0.78–4.96	0.78–1.93
Pork chop	18	LCSMs	B 4.34 ± 0.87 a	1.30–7.74	0.48–2.58	0.48–1.88
	16	SIOMs	A 6.21 ± 1.19 a	2.00–8.07	0.48–3.58	0.48–0.56

* In the level of aerobic mesophiles within the same sample type, means preceded by the same uppercase letter are not significantly different (*p* > 0.05); in the same column within the same store source and same microorganism, means followed by the same lowercase letter are not significantly different (*p* > 0.05); samples are obtained from 9 LCSMs and 10 SIOMs, respectively.

**Table 6 pathogens-12-00965-t006:** Occurrence of bacterial contamination in meat products procured from large chain supermarkets (LCSMs) and small, independently owned corner markets and convenience stores (SIOMs) *.

Sample Type	Sample Number	Market Source	Prevalence (%) of Positive Samples			
			*E. coli*	*Campylobacter*	*Listeria* spp.	*Salmonella*
Bacon	16	LCSMs	ND ^#^	ND	6.25	ND
	10	SIOMs	10	10	ND	ND
Beef cut	4	LCSMs	ND	ND	ND	ND
	8	SIOMs	12.5	ND	ND	ND
Ground beef	20	LCSMs	5	ND	ND	ND
	22	SIOMs	36	4.5	4.5	ND
Chicken leg	18	LCSMs	5.5	5.5	5.5	ND
	16	SIOMs	18.8	6.3	ND	6.3
Chicken wing	16	LCSMs	25	ND	ND	ND
	16	SIOMs	18.8	12.5	18.8	6.3
Pork chop	18	LCSMs	11.1	ND	11.1	ND
	16	SIOMs	12.5	6.3	ND	ND
Pork sausage	22	LCSMs	50	4.5	9	4.5
	18	SIOMs	50	ND	11.1	16.6

* Samples are obtained from 9 LCSMs and 10 SIOMs, respectively; ^#^ not detected.

**Table 7 pathogens-12-00965-t007:** Level of aerobic mesophilic counts on the packaging surfaces of samples procured from large chain supermarkets (LCSMs) and small, independently owned corner markets and convenience stores (SIOMs) *.

Sample Packaging	Sample Number	Store Source	Microorganism Population		
			Aerobic Mesophiles (log CFU/cm^3^)		Coliforms (log MPN/cm^3^)
			Average	Range	Range
Hot dog	12	LCSMs	A 2.28 ± 1.34 a	1.30–4.62	0.78–0.78
	12	SIOMs	A 1.55 ± 0.39 b	1.30–2.38	0.78–0.78
Sausage	22	LCSMs	B 2.07 ± 1.14 a	1.30–5.53	0.78–1.26
	15	SIOMs	A 3.07 ± 1.62 a	1.30–7.31	0.78–2.68

* In the level of aerobic mesophiles within the same sample packaging, means preceded by the same uppercase letter are not significantly different (*p* > 0.05); in the same column within the same store source and microorganism, means followed by the same lowercase letter are not significantly different (*p* > 0.05); samples are obtained from 9 LCSMs and 9 SIOMs, respectively.

**Table 8 pathogens-12-00965-t008:** Representative blendable ethnic food products procured from eight small, independently owned corner markets and convenience stores (SIOMs) and their bacterial prevalence *.

Sample Type	Sample Number (n = 45)	Microorganisms			Prevalence (%) of Bacteria		
		Aerobic Mesophiles	Coliforms	*E. coli*	*Campylobacter*	*Listeria* spp.	*Salmonella*
		Average	Average	Average			
Fat back	4	3.07 ± 0.71 c	0.48 ± 0.00 d	0.48 ± 0.00 b	0	0	0
Meat seasoned	2	8.23 ± 1.1 ab	1.16 ± 0.28 cd	0.92 ± 0.62 ab	0	0	0
Pork, jowl	4	5.33 ± 2.44 bc	0.50 ± 0.04 d	0.48 ± 0.00 b	0	0	0
Pollock, fillets	2	2.50 ± 0.71 c	0.48 ± 0.00 d	0.48 ± 0.00 b	0	0	0
Seafood salad	2	6.33 ± 0.04 ab	1.47 ± 0.71 c	0.48 ± 0.00 b	0	0	0
Shrimp, ground	2	5.38 ± 0.25 bc	0.48 ± 0.00 d	0.48 ± 0.00 b	0	0	0
Egusi, ground	2	5.36 ± 0.25 bc	3.18 ± 1.12 b	1.18 ± 0.00 ab	0	0	0
Okazi leave, dried	2	8.73 ± 1.8 a	5.04 ± 0.00 a	1.43 ± 1.34 a	100	0	100
Saffron, powder	2	7.72 ± 0.27 ab	0.52 ± 0.06 d	0.48 ± 0.00 b	0	0	0

* In the same column within each microorganism, means followed by the same lowercase letter are not significantly different (*p* > 0.05).

**Table 9 pathogens-12-00965-t009:** Representative non-blendable ethnic food products procured from eight small, independently owned corner markets and convenience stores (SIOMs) and their bacterial prevalence *.

Sample Type	Sample Number (*n* = 45)	Microorganisms		
		Aerobic Mesophiles	Coliforms	*E. coli*
		Average	Average	Average
Chicken, free range	2	3.53 ± 1.05 cd	1.13 ± 0.49 c	0.78 ± 0.00 a
Chicken, whole	2	8.05 ± 0.02 a	4.22 ± 1.05 a	0.86 ± 0.00 a
Cow, feet	2	2.93 ± 0.83 def	0.78 ± 0.00 c	0.78 ± 0.00 a
Goat meat	3	3.41 ± 0.40 cde	0.74 ± 0.24 c	0.58 ± 0.17 b
Lamb meat	2	3.71 ± 0.13 cd	0.48 ± 0.00 c	0.48 ± 0.00 b
Ox tail	2	4.30 ± 0.28 c	1.97 ± 0.43 b	0.78 ± 0.00 a
Pork, BBQ	2	2.39 ± 0.55 ef	0.48 ± 0.00 c	0.48 ± 0.00 b
Anchovies, smoked	2	8.33 ± 0.16 a	0.48 ± 0.00 c	0.48 ± 0.00 b
Shrimp, smoked	2	5.89 ± 0.41 b	0.48 ± 0.00 c	0.48 ± 0.00 b
Stock fish bites	2	4.44 ± 0.33 c	0.78 ± 0.00 c	0.78 ± 0.00 a
Date	2	2.00 ± 0.00 f	0.48 ± 0.00 c	0.48 ± 0.00 b

* In the same column within each microorganism, means followed by the same lowercase letter are not significantly different (*p* > 0.05).

**Table 10 pathogens-12-00965-t010:** Antimicrobial prevalence of 93 *E. coli*, 14 *Campylobacter*, 15 *Listeria*, and 8 *Salmonella* isolates in food samples procured from large chain super markets (LCSMs) and small, independently owned corner markets and convenience stores (SIOMs) between August 2018 and March 2020 *.

Bacteria	Nature of AMR ^a^	Market Source (*n*) ^b^	Prevalence (%) of Resistance or Non-Susceptibility to Each Quantity of Antimicrobial Agents ^c^											
			1	2	3	4	5	6	7	8	9	10	11	MDR (≥3) ^d^
*Campylobacter*	R	LCSMs (2)	0.0	0.0	0.0	0.0	0.0	50.0	50.0	0.0	0.0	0.0	0.0	100
		SIOMs (12)	0.0	0.0	25.0	16.7	25.0	0.0	25.0	0.0	0.0	0.0	0.0	66.7
		Total (14)	0.0	0.0	21.4	7.1	28.6	7.1	28.6	0.0	0.0	0.0	0.0	71.4
	R+I	LCSMs (2)	0.0	0.0	0.0	0.0	0.0	0.0	0.0	0.0	50.0	50.0	0.0	NA ^e^
		SIOMs (12)	0.0	0.0	8.3	0.0	0.0	16.7	16.7	33.3	8.3	8.3	8.3	NA
		Total (14)	0.0	0.0	7.1	0.0	0.0	14.3	14.3	35.7	14.3	7.1	7.1	NA
*E. coli*	R	LCSMs (33)	39.4	15.2	18.2	6.1	0.0	0.0	3.0	0.0	0.0	0.0	0.0	27.3
		SIOMs (60)	18.3	26.7	11.7	8.3	10.0	1.7	3.3	0.0	0.0	0.0	0.0	35.0
		Total (93)	25.8	22.6	14.0	7.5	6.5	2.2	2.2	0.0	0.0	0.0	0.0	32.3
	R+I	LCSMs (33)	30.3	21.2	15.2	9.1	3.0	3.0	3.0	0.0	0.0	0.0	0.0	NA
		SIOMs (60)	10.0	20.0	20.0	10.0	8.3	10.0	3.3	0.0	0.0	0.0	0.0	NA
		Total (93)	17.2	20.4	18.3	9.7	4.3	9.7	3.2	0.0	0.0	0.0	0.0	NA
*Listeria* spp.	R	LCSMs (9)	77.8	11.1	11.1	0.0	0.0	0.0	0.0	0.0	0.0	0.0	0.0	0.0
		SIOMs (6)	66.7	0.0	16.7	16.7	0.0	0.0	0.0	0.0	0.0	0.0	0.0	33.4
		Total (15)	73.3	6.7	13.3	6.7	0.0	0.0	0.0	0.0	0.0	0.0	0.0	13.3
	R+I	LCSMs (9)	77.8	0.0	0.0	0.0	11.1	11.1	0.0	0.0	0.0	0.0	0.0	NA
		SIOMs (6)	66.7	0.0	0.0	16.7	16.7	0.0	0.0	0.0	0.0	0.0	0.0	NA
		Total (15)	73.3	0.0	0.0	6.7	13.3	6.7	0.0	0.0	0.0	0.0	0.0	NA
*Salmonella*	R	LCSMs (1)	0.0	100	0.0	0.0	0.0	0.0	0.0	0.0	0.0	0.0	0.0	0.0
		SIOMs (7)	42.9	14.3	0.0	14.3	0.0	0.0	0.0	0.0	0.0	0.0	0.0	14.3
		Total (8)	37.5	25.0	0.0	12.5	0.0	0.0	0.0	0.0	0.0	0.0	0.0	12.5
	R+I	LCSMs (1)	0.0	100	0.0	0.0	0.0	0.0	0.0	0.0	0.0	0.0	0.0	NA
		SIOMs (7)	28.6	28.6	0.0	14.3	0.0	0.0	0.0	0.0	0.0	0.0	0.0	NA
		Total (8)	25.0	37.5	0.0	12.5	0.0	0.0	0.0	0.0	0.0	0.0	0.0	NA

* Susceptibility categorization was carried out in accordance with interpretive criteria provided by the National Committee of Clinical Laboratory Standards recommendations (CLSI 2015); ^a^ antimicrobial resistance (AMR); R: resistant; I: intermediate; R+I: non-susceptible to antimicrobial agents tested; ^b^ number of isolates tested; ^c^ prevalence (%) was presented in resistance and non-susceptibility of isolates to the total number of antimicrobial agents tested [i.e., an isolate exhibiting resistant and intermediate, respectively, to two and four antimicrobial agents was presented under 2 of Resistance and 6 of Non- susceptibility (R+I).]; ^d^ multidrug resistance; ^e^ not applicable.

## Data Availability

The data that support the findings of this study are available in the main manuscript.

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
