# Peer review of "Microbiological Survey and Antimicrobial Resistance of Foodborne Bacteria in Select Meat Products and Ethnic Food Products Procured from Food Desert Retail Outlets in Central Virginia, USA"

_pathogens, 2023, doi:10.3390/pathogens12070965_

Round 1
Reviewer 1 Report
The current study is of some interest as it combines social aspects with microbiological methods. Based on the description of the microbiological techniques, the studies were carried out carefully and according to accepted standards.
The writing is generally very good, but I question the placement of some of the Tables in the Materials and Method section. They should be in the Results section.
Below are some questions and suggestions for improving the wording of some sentences.
L. 17: A total of 326 samples??? {I think the type(s) of samples should be stated in the abstract}
L. 65: in the prevalence of??? [delete?] and AMR
Table 1: It is confusing to find a Table with results presented in the Method section. What is further confusing is the statement that some data are from a prior study without indicating which of the data are new.
Table 2: Again, results are shown in the Method section.
There does not appear to be a Table 3.
Table 4 and Table 6: “mesophiles” is misspelled.
L. 206: The colonies of presumptive Listeria on OL agar were identified to the species level ….
L. 211: presumptive instead or assumptive?
L. 228: and 231: It is better to provide the values as x g rather than rpm.
L. 303: “VBDH, MCDH, and SDDH checklists”: spelled out somewhere?
L. 367: SIOM was about the same except for a few commodities, …
L. 374 and 447: “ground beef”
L. 404: …., had??[was?] the highest.
L. 483: …much higher than that of the food desert-acquired meat products.
L. 519: On none of the packaging surfaces analyzed in the current study were E. coli, Listeria, and Salmonella detected (Table 7).
L. 531: ….samples of dried okazi leave had significantly ….
L. 534: …seasoned meat, smoked anchovies ….
L. 535: ….origin exhibited aerobic mesophile ….
L. 541 … are presented in the ….
L. 544 and 548: dried okazi leaves
L. 549: Listeria welshimeri was detected on the packaging surface of one ground egusi sample (data not shown).
Table 8: Either you use the terms such as “free-range chicken” or you add a comma between “chicken” and “free-range”. Same applies to other commodities in the Table.
L. 568: …most common (100%), followed …..
Table 9: I have a difficult time understanding this Table. What does “Prevalence (%) of resistance or non-susceptibility to each quantity of antimicrobial agents” mean? What is meant by “quantity”? Concnetration?
L. 585: The two Campylobacter ….
L. 649: In contrast, 100% of the isolates were susceptible to MEM, AMK, TOB, and CIP.
L. 664: Overall, the current study revealed a higher??[lower?] prevalence of MDR in the isolates …
L. 759: However, none of the bacterial species assessed in the current study showed 100% susceptibility to all antimicrobials.
L. 775: …..in all the tested all samples, ….
Please see attached file.
Author Response
The authors appreciate the thoughtful suggestions provided by the reviewer for the matter of revision. Following are our responses to the reviewer’s comments. We have made most of the changes suggested by the reviewers, and a list of our itemized responses to the reviewers is addressed below.
Below are some questions and suggestions for improving the wording of some sentences.
L. 17: A total of 326 samples??? {I think the type(s) of samples should be stated in the abstract}
As suggested by the reviewer, due to the limit of word counts in the Abstract section, some sample types used for the study were added.
L. 65: in the prevalence of??? [delete?] and AMR
The sentence was revised for clarification.
Table 1: It is confusing to find a Table with results presented in the Method section. What is further confusing is the statement that some data are from a prior study without indicating which of the data are new.
The authors submitted the manuscripts, Tables, and Figures separately to the MDPI journal. The authors believe that the manuscript format used by the reviewers is MDPI system-generated. However, the authors reorganized and placed Tables in the appropriate sections to ensure everything was clear. In addition, the study was conducted for three years and generated immense data with many Tables and Figures. Therefore, the authors decided to prepare two manuscripts from the findings, with one manuscript focusing on meat products and ethnic food products and the second on fresh produce only. Neither the manuscript nor any parts of its content are published or considered in another journal. Therefore, the authors decided to remove the statement “a portion of the compliance rate shown here is reported in a prior study (Kim et al. 2023)” from the text. Contents are revised accordingly, as well.
Table 2: Again, results are shown in the Method section.
There does not appear to be a Table 3.
Table 3 is “a list of antimicrobials and interpretive criteria used in this study.” and Tables are re-numbered accordingly.
Table 4 and Table 6: “mesophiles” is misspelled.
Revised as indicated.
L. 206: The colonies of presumptive Listeria on OL agar were identified to the species level ….
Revised as suggested.
L. 211: presumptive instead or assumptive?
Revised as suggested.
L. 228: and 231: It is better to provide the values as x g rather than rpm.
Revised as suggested.
L. 303: “VBDH, MCDH, and SDDH checklists”: spelled out somewhere?
Yes. They were spelled out in the footnote in Table 2.
L. 367: SIOM was about the same except for a few commodities…
Revised as suggested.
L. 374 and 447: “ground beef”
Revised as suggested throughout the manuscript.
L. 404: …., had??[was?] the highest.
Revised as suggested.
L. 483: …much higher than that of the food desert-acquired meat products.
Revised as suggested.
L. 519: On none of the packaging surfaces analyzed in the current study were E. coli, Listeria, and Salmonella detected (Table 7).
Revised as suggested.
L. 531: ….samples of dried okazi leave had significantly ….
Revised as suggested.
L. 534: …seasoned meat, smoked anchovies ….
Revised as suggested.
L. 535: ….origin exhibited aerobic mesophile ….
Revised as suggested.
L. 541 … are presented in the ….
Revised as suggested.
L. 544 and 548: dried okazi leaves
Revised as suggested.
L. 549: Listeria welshimeri was detected on the packaging surface of one ground egusi sample (data not shown).
Revised as suggested.
Table 8: Either you use the terms such as “free-range chicken” or you add a comma between “chicken” and “free-range”. Same applies to other commodities in the Table.
Revised as suggested.
L. 568: …most common (100%), followed …..
Revised as suggested.
Table 9: I have a difficult time understanding this Table. What does “Prevalence (%) of resistance or non-susceptibility to each quantity of antimicrobial agents” mean? What is meant by “quantity”? Concnetration?
As illustrated in the footnote, the prevalence of resistance means the percentage of bacteria showing resistance to the number (quantities) of antimicrobial agents tested. For example, 1 (50%) out of 2 Campylobacter isolated from the LCSM showed resistance to 6 different antimicrobial agents (yellow highlighted in the Table).
L. 585: The two Campylobacter ….
Revised as suggested.
L. 649: In contrast, 100% of the isolates were susceptible to MEM, AMK, TOB, and CIP.
Revised as suggested.
L. 664: Overall, the current study revealed a higher??[lower?] prevalence of MDR in the isolates …
Revised to “lower.”
L. 759: However, none of the bacterial species assessed in the current study showed 100% susceptibility to all antimicrobials.
Revised as suggested.
L. 775: …..in all the tested all samples, ….
Revised as suggested.
We feel that these changes have improved the manuscript and trust that you will let us know if anything else is required. Thank you very much for your help.
Sincerely,
Reviewer 2 Report
Dear Authors,
This manuscript assesses the levels of aerobic mesophiles, coliforms and E. coli, the occurrence of foodborne pathogens and prevalence of AMR bacteria in samples of selected meat products, ethnic food products, sold at both SIOM and LCSM, in the surrounding food desert areas of Petersburg and Colonial Heights, VA, USA.
The manuscript comprises the title, abstract, keywords, introduction, material and methods, results and discussion, conclusion, 51 references, 9 tables and 3 figures. However, some issues must be addressed before considering for publication, as it is difficult to read and understand due to some incoherences in the sections of “Material and methods” and “Results and discussion”. Also, it remains unclear if all the results are original, as it is stated that “a portion of the compliance rate shown here is reported in a prior study (Kim et al. 2023)” in the note of table 1. The content of the manuscript needs to assure that “neither the manuscript nor any parts of its content are ... published in another journal”.
In section “Material and methods”, first paragraph of subsection “Microbial Testing”, there is a lack of a comprehensive description of the tested samples, as the reader is referred to tables with results (Tables 4, 5, 6) and are mentioned “Sections I, II and III” which are not further mentioned.
In section “Results and discussion” the same results for E. coli are presented twice: as levels (tables 4, 6) and occurrence (tables 5, 7). Moreover, some data is presented in an unclear way. For example, E. coli levels and ranges are presented for packaging surface samples (table 6) where no E. coli was detetected (table 7). Particularly for coliforms and E. coli, it is incomprehensible how data analysis (averages, ranges, significance of differences) was performed for parameters with values below the quantification level. Besides, several non-blendable samples (such as, chicken legs and wings, cow feet, etc), were tested by inoculating the wash fluid (PW), so results expressed as log CFU are per g of wash fluid and not of sample. Consequently, the respective levels of aerobic mesophiles per mass unit of wash fluid should not be evaluated according to ICMSF (1986) limits (Tables 4 and 8), which are presented per mass unit of food. Figure 6 (A and B) is missing, so it is not possible to assess all the results/discussion.
In general, avoid using full stops in subtitles, renumber as 4 section “Conclusion”, correct the designations of mesophiles in tables 4, 6 and 8, renumber Figures 3 to 6 (as no Figures 1 to 2 exist), include references for the educational materials used and adapted (e.g., VBDH 2019, MCDH 2018, SDDH 2013), verify if all references are in-text cited (no citation was found for reference: Fett et al, 2014) and follow the alphabetic order for all references (Caspi et al, 2017 appears before Block and Kouba, 2008 in references).
Please see below for specific comments:
2. Materials and Methods
Tables with results (Tables 1, 2, 4, 5, 6) should be moved to “Results and discussion” section.
2.1. Food Safety Inspection
Please discriminate the practices evaluated for food storage and employees. How did you assess “ consumer acceptability”?
2.3. Microbial testing
Lines 135-144: Please rewrite paragraph one, without stating the non existing sections I, II and II, designate all the types of tested samples and specify all the relevant information (such as, identifying all the non-blendale samples) and discriminating among food itens, seasoning, etc. Perhaps, you could include a table or a figure without any result.
Line 178: What do you mean by equal dimension (cm3) of sterile PW”? Perhaps equal volume? Please rephrase.
Lines 181-185: It is important to include for each parameter, the units for expression of results per mass or volume of the tested matrix (food/seasoning, surface or wash fluid) and also include the tested matrix in the detection limit.
Lines 200-201: Please rephrase, as Campylobacter, Listeria and Salmonella are genus names not species.
3. Results and discussion
In table 4, the presentation of aerobic mesophile results with the assessment of samples according to ICMSF seems out of context. Instead, please include in table 4 the range of values aerobic mesophile, as already presented for coliforms and E. coli. The assessment of blendable samples according to ICMSF can be included in the text.
Please delete the prevalence of E. coli from tables 5 and 7; you can include this information in tables 4 and 6 or mention in the text. Also revise and uniformize the “subject” of the prevalence % in table 8, as not all tested bacteria are pathogens (e.g. Listeria innocua, which can be included in the % of Listeria spp.)
Lines 546-547: Please clarify how and why you consider “saffron powder” as a ready-to-eat food.
Line 722: Replace “Figure 5A” by “Figure 6A”.
Line 738: It is Kilonzo-Nthenge et al., 2016 or Kilonzo-Nthenge et al., 2017?
4. Conclusion
Please revise as some sentences are not conclusions odf the study (e.g., first sentence).
Best Regards,
Reviewer
Author Response
The authors appreciate the thoughtful suggestions provided by the reviewer for the matter of revision. Following are our responses to the reviewer’s comments. We have made most of the changes suggested by the reviewers, and a list of our itemized responses to the reviewers is addressed below.
The manuscript comprises the title, abstract, keywords, introduction, material and methods, results and discussion, conclusion, 51 references, 9 tables and 3 figures. However, some issues must be addressed before considering for publication, as it is difficult to read and understand due to some incoherences in the sections of “Material and methods” and “Results and discussion”. Also, it remains unclear if all the results are original, as it is stated that “a portion of the compliance rate shown here is reported in a prior study (Kim et al. 2023)” in the note of table 1. The content of the manuscript needs to assure that “neither the manuscript nor any parts of its content are ... published in another journal”.
The study was conducted for three years and generated immense data with many Tables and Figures. Therefore, the authors decided to prepare two manuscripts from the findings, with one focusing on meat and ethnic food products (the current manuscript) and the second on fresh produce only. Neither the manuscript nor any parts of its content are published or considered in another journal, the authors, therefore, removed the statement “a portion of the compliance rate shown here is reported in a prior study (Kim et al. 2023)” from the text. The second manuscript, focusing on fresh produce only, will be submitted with appropriate modifications to avoid publication of the same information.
In section “Material and methods”, first paragraph of subsection “Microbial Testing”, there is a lack of a comprehensive description of the tested samples, as the reader is referred to tables with results (Tables 4, 5, 6) and are mentioned “Sections I, II and III” which are not further mentioned.
As suggested by the reviewer, a comprehensive description of the tested samples for each section is included.
In section “Results and discussion” the same results for E. coli are presented twice: as levels (tables 4, 6) and occurrence (tables 5, 7). Moreover, some data is presented in an unclear way. For example, E. coli levels and ranges are presented for packaging surface samples (table 6) where no E. coli was detetected (table 7).
The authors agree with the reviewer. Therefore, the average data for coliforms and E. coli are removed, and the range of coliforms and E. coli population was presented in Tables 4 and 5 (with new numbering). However, the authors agreed to keep the occurrence of E. coli in Table 5 (Table 4 before the revision). Because the E. coli data presented in the Table 5 will manifest the prevalence of E. coli contamination in samples tested.
Particularly for coliforms and E. coli, it is incomprehensible how data analysis (averages, ranges, significance of differences) was performed for parameters with values below the quantification level.
For the data with values at or below the quantification/detection (0.48 log MPN/g) level, the authors used the detection level for statistical analysis purposes. However, considering the reviewer’s comment and avoiding confusion, the authors agreed to remove average columns for coliforms and E. coli.
Besides, several non-blendable samples (such as, chicken legs and wings, cow feet, etc), were tested by inoculating the wash fluid (PW), so results expressed as log CFU are per g of wash fluid and not of sample. Consequently, the respective levels of aerobic mesophiles per mass unit of wash fluid should not be evaluated according to ICMSF (1986) limits (Tables 4 and 8), which are presented per mass unit of food. Figure 6 (A and B) is missing, so it is not possible to assess all the results/discussion.
As suggested by the reviewer, the Tables were revised accordingly.
In general, avoid using full stops in subtitles, renumber as 4 section “Conclusion”, correct the designations of mesophiles in tables 4, 6 and 8, renumber Figures 3 to 6 (as no Figures 1 to 2 exist), include references for the educational materials used and adapted (e.g., VBDH 2019, MCDH 2018, SDDH 2013), verify if all references are in-text cited (no citation was found for reference: Fett et al, 2014) and follow the alphabetic order for all references (Caspi et al, 2017 appears before Block and Kouba, 2008 in references).
As suggested by the reviewer, appropriate corrections were made.
Please see below for specific comments:
- Materials and Methods
Tables with results (Tables 1, 2, 4, 5, 6) should be moved to “Results and discussion” section.
As suggested by the reviewer, Tables were moved to appropriate section.
2.1. Food Safety Inspection
Please discriminate the practices evaluated for food storage and employees. How did you assess “ consumer acceptability”? As requested by the reviewer, detailed discrimination is included in the section, and the Table was revised accordingly. In addition, to avoid unnecessary confusion, the text “consumer acceptability” was removed.
2.3. Microbial testing
Lines 135-144: Please rewrite paragraph one, without stating the non existing sections I, II and II, designate all the types of tested samples and specify all the relevant information (such as, identifying all the non-blendale samples) and discriminating among food itens, seasoning, etc. Perhaps, you could include a table or a figure without any result.
As suggested by the reviewer, the paragraph was rewritten.
Line 178: What do you mean by equal dimension (cm3) of sterile PW”? Perhaps equal volume? Please rephrase.
As inquired by the reviewer, the sentence was rephrased for clarification.
Lines 181-185: It is important to include for each parameter, the units for expression of results per mass or volume of the tested matrix (food/seasoning, surface or wash fluid) and also include the tested matrix in the detection limit.
As requested by the reviewer, the tested matrix and its detection limit is included in the text.
Lines 200-201: Please rephrase, as Campylobacter, Listeria and Salmonella are genus names not species.
As suggested by the reviewer, the correction was made.
- Results and discussion
In table 4, the presentation of aerobic mesophile results with the assessment of samples according to ICMSF seems out of context. Instead, please include in table 4 the range of values aerobic mesophile, as already presented for coliforms and E. coli. The assessment of blendable samples according to ICMSF can be included in the text.
As suggested by the reviewer, the correction was made.
Please delete the prevalence of E. coli from tables 5 and 7; you can include this information in tables 4 and 6 or mention in the text. Also revise and uniformize the “subject” of the prevalence % in table 8, as not all tested bacteria are pathogens (e.g. Listeria innocua, which can be included in the % of Listeria spp.)
As suggested by the reviewer, the corrections were made.
Lines 546-547: Please clarify how and why you consider “saffron powder” as a ready-to-eat food.
For the clarification, the saffron powder was removed from the ready-to-eat food.
Line 722: Replace “Figure 5A” by “Figure 6A”.
As suggested by the reviewer, the corrections were made.
Line 738: It is Kilonzo-Nthenge et al., 2016 or Kilonzo-Nthenge et al., 2017?
The correct citation is provided.
- Conclusion
Please revise as some sentences are not conclusions odf the study (e.g., first sentence).
As suggested by the reviewer, the revision was made.
We feel that these changes have improved the manuscript and trust that you will let us know if anything else is required. Thank you very much for your help.
Sincerely,
Reviewer 3 Report
The paper is devoted to study of microbial contamination of food products taken either in supermarkets or in small shops for low income citizens, which are called food deserts. The topic of the paper corresponds to a journal specialized either in food or in social problems, but not for Pathogens. Bacteria that are described as pathogenic quite can be non-pathogenic because they are described up to the genus. Meantime, only the genus Listeria includes 2 pathogenic and 18 non-pathogenic species. The same can be attributed to Campylobacter. Salmonella and E. coli should be characterized up to a serotype to be interesting for microbiologists working with pathogens. Bacterial contaminants are characterized using standard FDA methods and the total study resembles a routine practice of local microbiological labs monitoring marker places. Despite using standard methods, in Materials and Methods the authors reference to their previous publications that looks at least strange.
Author Response
The authors appreciate the thoughtful suggestions provided by the reviewer for the matter of revision. Following are our responses to the reviewer’s comments. We have made most of the changes suggested by the reviewers, and a list of our itemized responses to the reviewers is addressed below.
The paper is devoted to study of microbial contamination of food products taken either in supermarkets or in small shops for low income citizens, which are called food deserts. The topic of the paper corresponds to a journal specialized either in food or in social problems, but not for Pathogens. Bacteria that are described as pathogenic quite can be non-pathogenic because they are described up to the genus. Meantime, only the genus Listeria includes 2 pathogenic and 18 non-pathogenic species. The same can be attributed to Campylobacter. Salmonella and E. coli should be characterized up to a serotype to be interesting for microbiologists working with pathogens.
The authors agree with the comments made by the reviewer. However, although two Listeria only were pathogenic L. monocytogenes, reporting the detection of pathogenic Listeria and their AMR prevalence from food samples sold in the studied area may be interesting for microbiologists working with pathogens. In addition, the PCR method used for Campylobacter confirmation was based on the pathogenic C. jejuni- and C. fetus-specific real-time PCR assay. Therefore, the Campylobacter we isolated may be highly related to pathogenic Campylobacter. API 20E test profile (6704552) also indicated four strains of Salmonella very similar to the profiles of Salmonella enterica. Therefore, the authors believe that along with the findings, reporting their prevalence in AMR is aligned with the specialized journal. Furthermore, molecular analysis is being currently conducted to characterize and determine genes responsible for antimicrobial resistance and virulence of thousands of Campylobacter, E. coli, Listeria, and Salmonella isolates obtained from various environmental and food samples. However, considering the reviewer’s comment, the authors decided to revise the title to “Microbiological Survey and Antimicrobial Resistance of Foodborne Bacteria in Select Meat Products and Ethnic Food Products Procured from Food Desert Retail Outlets in Central Virginia, USA.”
Bacterial contaminants are characterized using standard FDA methods and the total study resembles a routine practice of local microbiological labs monitoring marker places. Despite using standard methods, in Materials and Methods the authors reference to their previous publications that looks at least strange.
Considering the reviewer’s inquiry, the authors’ references cited were removed.
We feel that these changes have improved the manuscript and trust that you will let us know if anything else is required. Thank you very much for your help.
Sincerely,
Round 2
Reviewer 2 Report
Dear Authors,
The issues and suggestions have been addressed and changes have been made accordingly, so that the manuscript is improved and now can be recommended for publication.
Best Regards,
Reviewer
Reviewer 3 Report
Authours signficantly improved the manuscript.